# Dynamic Immune Response to Vibriosis in Pacific Oyster *Crassostrea gigas* Larvae during the Infection Process as Supported by Accurate Positioning of GFP-Tagged *Vibrio* Strains

**DOI:** 10.3390/microorganisms9071523

**Published:** 2021-07-17

**Authors:** Dongdong Wang, Alfredo Loor, Lobke De Bels, Gilbert Van Stappen, Wim Van den Broeck, Nancy Nevejan

**Affiliations:** 1Laboratory of Aquaculture & *Artemia* Reference Center, Department of Animal Sciences and Aquatic Ecology, Faculty of Bioscience Engineering, Ghent University, 9000 Ghent, Belgium; Alfredo.Loor@UGent.be (A.L.); Gilbert.VanStappen@UGent.be (G.V.S.); Nancy.Nevejan@UGent.be (N.N.); 2Department of Morphology, Faculty of Veterinary Medicine, Ghent University, Salisburylaan 133, 9820 Merelbeke, Belgium; Lobke.DeBels@UGent.be (L.D.B.); Wim.VandenBroeck@UGent.be (W.V.d.B.)

**Keywords:** *Crassostrea gigas* larvae, vibriosis, pathogenesis, GFP, histopathology, immune system

## Abstract

As the immune system is not fully developed during the larval stage, hatchery culture of bivalve larvae is characterized by frequent mass mortality caused by bacterial pathogens, especially *Vibrio* spp. However, the knowledge is limited to the pathogenesis of vibriosis in oyster larvae, while the immune response to pathogenic microorganisms in this early life stage is still far from being fully elucidated. In this study, we combined green fluorescent protein (GFP)-tagging, histological and transcriptomic analyses to clarify the pathogenesis of experimental vibriosis and the mechanisms used by the host Pacific oyster *Crassostrea gigas* larvae to resist infection. The *Vibrio* strains first colonized the digestive system and rapidly proliferated, while only the transcription level of IκB kinase (*IKK*) and nuclear factor κB (*NF-κ**B*) associated with signaling transduction were up-regulated in oyster at 18 h post challenge (hpc). The mRNA levels for *integrin β-1*, *peroxinectin*, and heat shock protein 70 (*HSP70*), which are associated with phagocytosis, cell adhesion, and cytoprotection, were not upregulated until 30 hpc when the necrosis already happened in the larval digestive system. This suggested that the immunity in the early stages of *C. gigas* is not strong enough to prevent vibriosis and future research may focus on the strengthening of the gastrointestinal immune ability to defend vibriosis in bivalve larvae.

## 1. Introduction

The Pacific oyster *Crassostrea gigas* is an important bivalve species with worldwide distribution and a 643,549 ton production in 2018, valued at US $1.36 million [1]. Bacteria of the genus *Vibrio* have been frequently documented as the main causative agent of diseases affecting hatchery-reared oyster larvae. The predominant species responsible for oyster larvae mortality outbreaks are *Vibrio splendidus*, *V. anguillarum*, *V. coralliilyticus*, *V. pectenicida*, *V. tubiashii*, and *V. aestuarianus* [2,3,4,5,6,7,8,9,10,11]. Numerous studies have reported approximately the same symptoms of bacillary necrosis regardless of the *Vibrio* species, characterized by a rapid onset of lower larval motility, detached velum, and necrotic soft tissue. However, there is limited knowledge on the pathogenesis of vibriosis in oyster larvae, while the immune response to pathogenic microorganisms in early life stages of oysters and even in marine invertebrates in general is still far from being fully elucidated.

The first phase in vibriosis of bivalve mollusks is attaching to and colonizing host tissues. Histological observation was first applied for exploring the colonization routes of *Vibrio* cells in bivalve larvae [12,13]. Histological sections of different bivalve mollusks larvae confirmed massive bacterial proliferation with extensive cellular destruction [12,13]. Necrosis and disorganization of tissue in *Vibrio*-infected carpet shell clam (*Ruditapes decussatus*) larvae [14] and in larvae of Eastern oyster *Crassostrea virginica* [15] were documented. Sandlund et al. 2007 [16] observed *V. pectenicida* inside the digestive area and mucosal cells of great scallop *Pecten maximus* larvae through immunohistochemical staining. Histopathological analyses revealed focal bacterial abscesses in the mantle tissue of *C. gigas* larvae [17] and necrosis and vacuolation in the digestive region of Greenshell^TM^ mussel larvae [18]. While *Vibrio* infection is a dynamic progression, histological analysis is limited to static destruction and does not allow distinguishing the tested vibrios from the regular microbiota inside the larvae. Hence, it is difficult to associate the clinical signs of vibriosis in larvae with the damage of larval tissues at the same infection time.

Green fluorescent tagging is a powerful tool, especially to track bacteria, without any other fluorescent probe or staining requirement. Therefore, the use of green fluorescent protein (GFP)-tagging in pathogenic *Vibrio* species has recently facilitated the unraveling of bacterial infection dynamics in vivo during infection [19,20,21,22]. Recently, the complete invasion pathway in bivalve larvae has been described by using the GFP-tagging tool. Dubert et al. 2016 [22] demonstrated that GFP-tagged *Vibrio* spp. firstly reached the digestive system of Manila clam (*Ruditapes philippinarum*) 10-day-old larvae before colonizing the whole animal. Nevertheless, the tiny size of the two-day-old *C. gigas* larvae (~90 μm in length) makes it impossible to explicitly separate larval organs and precisely localize the lesions under epifluorescence microscope. Therefore, in this study, we combined histopathological and GFP-tagging analyses to elaborate the bacterial entrance routes and the precise localization of bacillary necrosis in *C. gigas* larvae.

Furthermore, bivalves rely on an innate immunity system, which determines the immunological capacity of bivalve larvae during the ontogenesis [8,23,24,25,26,27]. Hemocytes are responsible for the cellular immune response, and phagocytosis is regarded as a critical process in the ontogeny of the bivalve immune system [28,29,30,31]. Moreover, various humoral molecules are released into the hemolymph, accompanied by the phagocytic process. When the infection occurs, pattern recognition receptors (PRRs) produced by hemocytes firstly recognize and combine the conserved pathogen-associated molecular patterns (PAMPs) on the surface of invaders, such as peptidoglycan (PGN), lipopolysaccharides (LPS), lipoteichoic acid (LTA), and β-1,3-glucans. Then the immune recognition may trigger different signaling pathways to induce downstream immune responses [32,33]. Previous studies on immune-related gene expression have indicated that both the cellular and humoral immune system appears to aid *C. gigas* larvae in its defense against bacterial infection during the early developmental stages, with immune recognition, signal transduction, apoptosis, and inflammatory response as main immune protection functions [8,24,34,35].

Song et al., 2016 [35] suggested that the immune system emerged earlier to aid larvae in defending bacterial infection during the early stages of *C. gigas* larval development. Moreover, the velum and digestive gland of oyster larvae, as the main entry of pathogens, were suggested to be the potential sites of the immune system [35]. Hence, besides observing the onset and advance of bacterial infection, it is also of high importance to document the immune response of oyster larvae through the infection process. Therefore, in addition to the combination of GFP-tagging and histological approach, as explained above, we also aimed to assess the dynamics of immune-related gene expression in oyster larvae, thus using a unique set of tools to understand the interaction between host and bacteria across the entire infection phases. In order to examine the modulation of immune genes in *C. gigas* larvae challenged with the pathogen, six immune related genes were selected. These genes have been documented to participate in various immune responses in bivalves and shrimp, such as immune elimination (superoxide dismutase (SOD), heat shock protein 70 (HSP70)) [36,37], cell adhesion (integrin β-1 and peroxinectin) [38,39,40,41], and signaling transduction (nuclear factor κB (NF-κB) and IκB kinase (IKK)) [42,43]. Two bacterial strains, *V. splendidus* ME9 and *V. anguillarum* NB10, that are pathogenic for blue mussel larvae (*Mytilus edulis*) were used in this study [44,45]. Moreover, some studies have reported that GFP expression in bacteria may confer a fitness cost, hence influence bacterial physiology [20,46,47]. To assess the possible effect on bacterial physiology, we compared the bacterial growth, virulence factors, and their toxicity towards *C. gigas* larvae before and after GFP-tagging.

## 2. Materials and Methods

### 2.1. Bacterial Strains and Culture Conditions

The two pathogenic strains *V. splendidus* strain CAIM 1923 (ME9) and *V. anguillarum* LMG 32357 (NB10) were stored at −80 °C in the Laboratory of Aquaculture & *Artemia* Reference Center (Ghent University). Strains were made rifampicin-resistant, as described in Wang et al. 2021 [45]. Their natural resistant mutants were stored at −80 °C.

For further experiments, strain ME9 and NB10 were cultured overnight in LB_35_ medium at 18 °C at 130 rpm. Cell density was adjusted at 550 nm using a spectrophotometer (Thermo Spectronic Genesis 20, Merelbeke, Belgium), setting the optical density (OD) values between 0.2–0.4. A conversion factor of OD of 1 equating to 1.2 × 10^9^ CFU mL^−1^ was applied.

### 2.2. DNA Transfer and Confirmation of GFP-Tagged Strains

GFP gene insertion was obtained by conjugation between the rifampicin-resistant strains ME9 and NB10 (recipient) and the donor *Escherichia coli* DH5α carrying the suicide plasmid pJBA120 containing a mini-Tn5 transposon with the *gfp* gene [48]. Succeeding selection, confirmation, and identification of GFP-tagged strains were performed as described in Wang et al., 2021 [45].

### 2.3. The Effect of GFP Incorporation on Cell Physiology of Bacteria

#### 2.3.1. Bacterial Growth Curve

ME9 and NB10 were grown in LB_35_ containing 50 μg mL^−1^ rifampicin at 23 °C on a rotary shaker at 120 rpm until 0.5 OD at 550 nm. Similarly, their GFP-tagged strains ME9-GFP and NB10-GFP were grown in LB_35_ containing 50 μg mL^−1^ rifampicin and 50 μg mL^−1^ kanamycin until 0.5 OD at 550 nm. The culture broth was inoculated (2% *v*/*v*) in 600 μL of LB_35_ containing 50 μg mL^−1^ rifampicin (for parents) or 50 μg mL^−1^ kanamycin and 50 μg mL^−1^ rifampicin (for transconjugants) in a 24-well plate in quadruplicate. The plate was then sealed with parafilm and incubated at 23 °C on a rotary shaker at 120 rpm. The growth of bacteria was monitored every two hours until 48 h by measuring OD of each well at 550 nm using a microplate reader (Tecan Infinite M200).

#### 2.3.2. Virulence Factors of the Vibrio Strains

To understand whether the GFP incorporation influences the virulence factors of the selected *Vibrio* strains ME9 and NB10, swimming motility was compared between the GFP-tagged strains and their parents, as well as lipase, phospholiphase, caseinase, gelatinase, and hemolysin activity, according to Natrah et al. 2011 and Yang and Defoirdt 2015 [49,50]. All assays were performed at least in triplicate.

For the swimming motility assay, 5 μL of overnight cultured *Vibrio* strains (OD_550_ = 1) was spotted in the center of soft LB_35_ plates (0.2%) [50]. The diameter of the motility zones was measured after 18 h of incubation at 23 °C.

As for the production of the lytic enzymes lipase, phospholipase, caseinase, gelatinase, and hemolysin, for each assay, overnight cultured *Vibrio* strains were diluted to an OD_550_ of 1 and 2 μL of the diluted cultures was spotted in the center of the test plates.

Lipase and phospholipase activity was assessed by adding 1% Tween 80 (Sigma-Aldrich) or 1% egg yolk emulsion (Sigma-Aldrich) to the LB_35_ agar, respectively. The diameter of the opalescent zones was measured after 3 days of incubation at 23 °C.

The caseinase assay plates were prepared by mixing equal volumes of a 4% skimmed milk powder suspension (Oxoid, Basingstoke, Hampshire, UK), sterilized at 121 °C for 5 min, and autoclaved double strength LB_35_ agar. Colony diameters and clearing zones were measured after 3 days of incubation at 23 °C. Gelatinase assay plates were prepared by mixing 0.5% gelatin (Sigma-Aldrich) with the LB_35_ agar. After 4 days of incubation, a saturated solution of ammonium sulfate (80%) in distilled water was poured over the plates and after 2 min, the diameter of the clearing zones around the colonies was measured. Hemolytic activity assay was performed by adding 5% defibrinated sheep blood (Oxoid, Basingstoke, Hampshire, UK) into autoclaved LB_35_ agar. Colony diameters and clearing zones were measured after 3 days of incubation at 23 °C.

### 2.4. Gnotobiotic C. gigas Larvae Challenge Test

#### 2.4.1. Rearing of *C. gigas* D-Veliger Larvae

To obtain axenic *C. gigas* D-veliger larvae, the following protocol was used, as modified from Langdon 1983 and Hung et al. 2015 [51,52]. Mature Pacific oysters obtained from the shellfish hatchery of Roem van Yerseke (Yerseke, The Netherlands) were cleaned under flowing seawater, and then sterilized externally for 30 s in a bath of 70% ethanol under a laminar flow. The shells were then washed with a commercial Betadine solution (diluted 50% with distilled water) for 5 min before opening them. The gonads were wiped with 0.5% hypochlorite solution and incised with a sterile scalpel, followed by recovering gametes by pipetting. Five mature males and five mature females were identified and sperms and eggs were separated and mixed in an individual beaker containing 500 mL sterile seawater at 23 °C. A controlled fertilization was obtained by gently mixing the eggs with an aliquot of sperm at a ratio of 10:1 in a beaker containing 1 L of filtered autoclaved sea water (FASW). After the polar bodies appeared in most of the embryos, the embryos were gently rinsed to remove excess sperms, and then transferred to 2 L glass bottles at a density of 100 eggs mL^−1^ at 23 °C with a mixture of the antibiotics chloramphenicol, nitrofurazone, and enrofloxacin (each at 10 mg L^−1^). After 2 days of incubation, D-veliger larvae were harvested on a sterile 50 µm sieve and washed gently with FASW at least five times to remove the antibiotics. Rinsed larvae were transferred to a sterile beaker filled with FASW and the density of larvae was adjusted to 200 larvae mL^−1^.

#### 2.4.2. General Design of the Challenge Tests

The challenge tests were conducted in 1 L Schott bottles. The first three experiments had a common set-up and compared the virulence between the two *Vibrio* strains ME9 and NB10, and between the GFP-tagged transconjugants and the respective parent strains by measuring the survival of oyster larvae (experiment 1). The infection process was followed under the phase contrast microscope and epifluorescence microscope (experiment 2), and histopathological analysis was carried out (experiment 3). In the next experiment, the expression of *C. gigas* immune-related genes was identified after exposing the larvae to the pathogen (experiment 4). The survival of larvae in each experiment was measured, but only the data from experiment 1 are shown here. The survival in other experiments shown in Appendix A was used to check if the toxicity of the tested strains remains stable on oyster larvae.

Each treatment was performed in triplicate. To quantify the microbial load in larvae, at the onset of each challenge test, a subsample of about 2000 rinsed larvae (cf. under Section 2.4.1) was homogenized in triplicate in 100 µL sterile sea water, and then plated on marine agar (MA). Moreover, in each challenge test, aliquots of 100 µL of the rearing water from each control treatment were plated on MA and TCBS once daily during the 4 days of challenge test. All the plates were incubated at 23 °C for two days. Only if no growth on any plate was observed, the data were used. Otherwise, the challenge test was repeated.

#### 2.4.3. Experiments 1, 2, and 3

The larvae were transferred to sterilized 1000 mL Schott bottles at a density of 100 larvae mL^−1^, with 500 mL FASW in each bottle supplemented with LB_35_ (0.1 % *v*/*v*) and rifampicin (10 mg L^−1^). The parental strains (ME9 and NB10) and their GFP-tagged strains (ME9-GFP and NB10-GFP) were respectively added into the rearing water at 10^6^ cells mL^−1^. Bottles with larvae to which no bacteria were added served as control treatment. Each treatment had 3 replicates and then the culture was incubated for 4 days at 23 °C.

In experiment 1, approximately 100 larvae were aseptically collected from a 10 mL subsample removed from each bottle using a micropipette at 24, 48, 72, and 96 h post introduction of the pathogens. Clinical signs were observed using a Nikon eclipse E200 microscope and pictures captured with a Nikon Digital Sight DS-Fi2 camera. Mortality was assessed as well by staining the larvae with lugol (5% *v*/*v*). Dark stained larvae were counted as alive, while the larvae were considered dead if only parts of the organs were stained or if shells were totally empty.

In experiment 2, in order to follow the whole infection process, approximately 200 larvae were aseptically collected from a 10 mL subsample, which was micropipetted from every bottle challenged with GFP-tagged strains, every 6 h for the first 36 h and every 12 h thereafter until 96 h. The larvae were monitored under phase contrast microscope and epifluorescence microscope to follow up the infection process (cf. Section 2.5).

To match sampling points of experiment 2, the larvae challenged with ME9-GFP and NB10-GFP in experiment 3 were sampled for histological analysis every 6 h for the first 36 h and every 12 h thereafter until 96 h (cf. Section 2.6). At each time point, approximately 4000 larvae were aseptically collected from each bottle.

#### 2.4.4. Experiment 4

The gene expression of *C. gigas* immune-related genes after being challenged with the pathogens was followed up in experiment 4. The larvae were transferred to sterilized 1000 mL Schott bottles at a density of 100 larvae mL^−1^. Each bottle contained 500 mL FASW supplemented with rifampicin (10 mg L^−1^) and LB_35_ (0.1 % *v*/*v*). Only ME9-GFP was used for the bacterial challenge at a concentration of 10^6^ cells mL^−1^ because the same infection routes for ME9-GFP and NB10-GFP were observed during experiment 2. Each treatment had 3 replicates and was incubated for 4 days at 23 °C. The larvae, either infected by ME9-GFP or not infected (control treatment), were sampled every 6 h until 30 h for gene expression analysis (cf. Section 2.6).

### 2.5. Analytical Work

#### 2.5.1. Observation of Infection Process

The larvae from the control treatment were applied to distinguish the larval autofluorescence before monitoring the GFP-fluorescence. Firstly, the clinical signs of bacillary necrosis in larvae were observed with an inverted microscope (Nikon TMS) in phase contrast. In parallel, the infection progress was monitored using an epifluorescence microscope (40×) (Zeiss Axioscope 2 Plus, Oberkochen, Germany) with filter FLUO (495 nm). At the point of capturing images for fluorescence, the larvae were anaesthetized using 15% MgCl_2_.

#### 2.5.2. H&E Stained Histological Analysis

Following Wang et al. 2021 [45], at each time point, approximately 4000 larvae were collected from each bottle and fixed in Davidson’s fixative for 24 h. The larvae were stored in 70% ethanol at 4 °C and then dehydrated in a series of 80%, 2× 94%, 2× 100% isopropanol, cleared in two series of xylene and later infiltrated in paraffin wax in a cassette to create a mounting block. Sections of 3 µm were cut using a microtome (Microm HM 360 Rotary Microtome, Walldorf, Germany) and then stained by using a Thermo Scientific Gemini AS Automated Slide Stainer (Thermo Fisher Scientific, Inc., Waltham, MA, USA). The slides were dipped in two series of xylene for 5 min each and two series of isopropanol for 5 min each, and then rehydrated through a series of 94%, 80%, 70%, and 50% ethanol, followed by distilled water. Slides were stained with haematoxylin and eosin (H & E), rinsed with distilled water, and then dipped in an ascending series of 50%, 70%, 80%, and 94% ethanol, followed by clearing in two series of isopropanol and two series of xylene. Finally, slides were mounted with DPX mountant (BDH, UK). The samples were viewed using an Olympus BX61 microscope at a magnification of 100× and images were captured using the Olympus DP73 camera.

#### 2.5.3. Modulation of Immune Genes in Bacterial Challenged Larvae

##### RNA Extraction and cDNA Synthesis

The larvae for RT-qPCR analyses were harvested, frozen in liquid nitrogen, and stored at −80 °C until analysis. Total RNA was extracted using Qiagen RNeasy Plus Mini Kit (Cat No. 74136) according to the manufacturer’s instructions. RNA quality and quantity were assessed by ND-1000 NanoDrop spectrophotometer (ThermoFisher Scientific, Merelbeke, Belgium). Reverse transcription was done from 1 µg of RNA samples with the RevertAid H Minus First Strand cDNA Synthesis Kit (ThermoFisher Scientific, Merelbeke, Belgium) according to the manufacturer’s instructions.

##### Immune-Related Genes and Primers Used in the Study

Specific primers for gene *SOD* (Table 1) were designed using Primer3Plus^TM^ (www.primer3plus.com (accessed on 7 July 2019)) with predicted product sizes in the range of 120–180 bp. As for other genes, the primer sequences were collected from previously published research (Table 1). *RS18* and *RL7*, which were previously identified as the most stable expressed reference genes in Pacific oyster *C**. gigas*, were used for qPCR normalization [53].

#### Quantitative Real-Time PCR (RT-qPCR) Analysis

Quantitative reverse transcriptase real-time PCR was performed with Maxima^®^ SYBR Green/ROX qPCR Master Mix (Fisher Scientific, Erembodegem, Belgium). The amplification was performed in a total volume of 15 µL, containing 7.5 µL of 2× SYBR green master mix, 4 µL of cDNA (10 ng) (cf. Section “RNA Extraction and cDNA Synthesis”), and 1.5 µL of each specific primer. A master mix was prepared for four biological replicates and two technical replicates for each sample. The thermal cycling consisted of an initial denaturation at 95 °C for 10 min followed by 40 cycles of denaturation at 95 °C for 15 s, primer annealing at 60 °C for 30 s and elongation at 72 °C for 30 s. Negative control reaction was included for each primer set by omitting template cDNA. Data acquisition was performed with the StepOne Software v2.3 (Applied Biosystems, Forster City, CA, USA). The comparative CT method (2^−ΔΔCt^ method) following Livak and Schmittgen 2001 [54] was used to analyze the expression level of the target genes.

### 2.6. Statistical Analysis

To study the effect of GFP incorporation on the growth of the considered *Vibrio* strains, a two-way repeated measures ANOVA was conducted, taking into account the replicate measurements, and using GFP labeling and time as factors. The effect of GFP labeling on the activities of virulent determinants was analyzed by one-way ANOVA, followed by Bonferroni post-test, except for the hemolytic activity that was analyzed by Student’s t-test. To identify significant differences between the control and the treatments, the survival data of *C. gigas* larvae were subjected to two-way repeated measures ANOVA, followed by Bonferroni post-test, with *Vibrio* strains and time as factors. Results for the gene expression were represented as fold-changes relative to the geometrical mean of two internal control genes (*RS18* and *RL7*). The expression level in the control group was regarded as 1.0 and the expression ratio of the oyster larvae challenged by ME9-GFP was expressed in relation to the control group. To identify significant differences between control and treatment, the expression of immune-related genes was analyzed by Student’s *t*-test using log-transformed 2^−ΔΔCt^ values. Statistical analyses were performed using the Statistical Package for the Social Sciences (SPSS) version 25.0 using a significance level of 5%.

## 3. Results

### 3.1. The Effect of GFP Incorporation on Bacterial Growth and Virulence Factors

The results showed that there were no differences in the growth of ME9-GFP and NB10-GFP, as compared with their parental strains (*p* < 0.01, two-way repeated measures ANOVA) (Figure 1). Moreover, ME9 displayed a faster growth than NB10, and this pattern was also true for their respective GFP-tagged strains.

The GFP expression was associated with decreased swimming motility of both *Vibrio* strains (*p* < 0.001, one-way ANOVA) (Figure 2A). No significant differences in activity of the lytic enzymes between the parental strains and their corresponding GFP-tagged strains were found (Figure 2B–E). The bacteria tested positive for all virulence factors considered, except for lipase activity, since no clearing zone was observed around colonies in this case (data not shown). In addition, hemolytic activity was only observed in the strains ME9 and ME9-GFP, while NB10 and NB10-GFP did not show this activity (Figure 2E).

### 3.2. Survival Rate of the Oyster Larvae Infected with Vibrio Strains

Following the designed challenge model, healthy two-day-old D-veliger larvae were challenged with the GFP-tagged pathogens and their parental strains in vivo, at a concentration of 10^6^ CFU mL^−1^. NB10 and NB10-GFP showed less than 10% mortality during the first two days, but significantly higher mortalities at day 4 as compared with the control treatment (*p* < 0.05, two-way repeated measures ANOVA) (Figure 3). ME9 and ME9-GFP caused significantly higher mortalities than NB10 from day 2 onwards. However, there was no significant survival difference at each time point between the GFP-tagged strain and the respective parental strain for the two species. At the end of the challenge test, ME9 and ME9-GFP caused more than 50% mortality at day 4, while NB10 and NB10-GFP caused ~30% larval death (Figure 3). The toxicities of strains ME9-GFP and NB10-GFP used in the following experiments remained stable on oyster larvae (Appendix A).

### 3.3. Infection Process and Synchronous Histological Changes in Oyster Larvae Challenged by GFP-Tagged Vibrio spp.

The invasion pathway of ME9-GFP was quite similar to that of NB10-GFP and for this reason, the latter is not shown. Generally, the larvae showed different infection phases in the same bottle and this asynchronous phenomenon persisted throughout the whole infection process. Thus, the infection phases appearing in the majority of larvae were recorded and grouped together through time.

A first spot lit up green fluorescent in the stomach area of the larvae at 6 hpc. This indicated that *Vibrio* spp. were filtered by larvae through the velum and entered the stomach via the esophagus (Figure 4C). The fluorescence started becoming brighter in the stomach from 6 hpc onwards until 24 hpc, illustrating that the bacteria colonized the digestive system from 6 hpc and proliferated rapidly during the first 24 h of infection (Phase Ⅰ: the incubation phase) (Figure 4C,F,I,L). No clinical signs of infection were noticed at 24 h. The larvae still showed a normal appearance with an active swimming pattern and stretchable velum with abundant cilia movements. However, the histopathological analysis showed that the necrosis of larval tissues started with disorganized cilia in the stomach at 12 hpc (Figure 4D) and continued with the fragmentation of the stomach and digestive glands at 24 hpc (Figure 4J).

During the rapid diffusion phase (Phase Ⅱ: 30–48 hpc), the *Vibrio* strain ME9-GFP extended to the surrounding organs localized in the ventral region (shell-opening side) (Figure 4O,R). Simultaneously, the loss of digestive organs in the dorsal region (umbo side) and velum detachment was gradually visualized as the infection was progressing. At 48 h of infection, the larvae showed an abnormal extended velum with disorganized cilia and destroyed soft organs in the dorsal region (Figure 4Q): a damaged stomach and style sac at 36 hpc (Figure 4M) and the loss of the whole stomach, digestive glands, and velum retractor muscle at 48 hpc (Figure 4P). In the meantime, the first signs of disease were observed under the inverted microscope, characterized by circular or erratic swimming behavior and the incapability of retracting the extended velum.

From 72 to 96 hpc, the infection entered into the acute mortality phase (Phase Ⅲ). At 72 hpc, degradation continued with the loss of additional velum mass via the release of velar cells into the surrounding seawater (Figure 4T). At 96 hpc, larval death was clearly observed by the loss of distinguishable internal structures and the presence of an empty, relatively colorless shell (Figure 4W,X).

### 3.4. The Effect of Vibrio Infection on Expression of Oyster Immune-Associated Genes

Over time, the six immune-related genes displayed different patterns. The transcriptional levels of the signaling transduction-related genes, *REL*/*NF-κB* and *IKK*, showed the first up-regulation after *Vibrio* challenge, with 2.5- and 1.4-fold, respectively, at 18 hpc as compared with the control treatment (*p* < 0.05, Student’s *t*-test) (Figure 5). As the infection was progressing, the expression levels of *REL*/*NF-κB*, *integrin β-1*, *peroxinectin* and *HSP70* were significantly up-regulated at 30 hpc, namely 2.0-, 2.0-, 5.7-, and 2.3-fold, respectively (*p* < 0.05, Student’s *t*-test). There was neither repression nor stimulation in *SOD* expression in our study.

## 4. Discussion

Recently, GFP has been used as a fluorescence marker in pathogenic *Vibrio* species to unravel bacterial infection dynamics in bivalve molluscs, shrimps, and fish [19,20,21,22,45,55,56,57]. However, it is recommended to assess the possible effect of GFP-tagging on bacterial physiology for every specific experimental set-up since GFP expression in bacteria may confer a fitness cost [20,46,47]. Hence, we compared the bacterial growth, virulence factors, and their virulence towards *C. gigas* larvae before and after GFP-tagging at 23 °C, the suitable temperature for culturing *C. gigas* larvae. The results showed that the only change due to GFP incorporation was the slower swimming motility, but this change did not result in lower bacterial virulence towards *C. gigas* larvae in vivo. The possible explanation could be that virulence in pathogenic *Vibrio* strains is multifactorial [58]. The associated factors are not restricted to motility but other factors should also be considered, such as adhesion [59], invasion [60], outer membrane protein [61,62], and other extracellular enzymes [63,64,65,66]. Hence, it would appear that the virulence of the strains ME9 and NB10 remains stable after the incorporation of GFP in our study.

Overall, there are four types of pathogenesis of experimental vibriosis in bivalve larvae reported by previous studies [13,45]. By using histological, immunofluorescent, and ultrastructural analyses, Elston and Leibovitz 1980 [13] described the first three types of pathogenesis of vibriosis in American oyster *C. virginica* larvae. In type Ⅰ pathogenesis, pathogenic *Vibrio* attaches to the larval shell and grows along the larval mantle causing progressive disruption of the mantle from the periphery inward. Subsequently, the bacteria infect the visceral cavity and eventually induce larval death. In type Ⅱ pathogenesis, velar deformation without *Vibrio* invasion into tissues is first noted, followed by complete necrosis of digestive organs and subsequent bacterial colonization within the visceral cavity as the disease progresses. In pathogenesis type Ⅲ, the vibriosis is characterized first by progressive and extensive visceral atrophy with shrunken visceral mass occupying only the dorsal region, and subsequently by the invasion of bacteria with focal lesions in organs of digestive tract. Wang et al. 2021 [45] recently described a new type of pathogenesis (type IV) of bivalve vibriosis by simultaneously using GFP-tagging, histopathology, and ultrastructural analyses in blue mussel (*M. edulis*) larvae. It was similar to pathogenesis type Ⅱ but occurred inversely: the pathogenic *Vibrio* grows along the digestive tract, but the velar abnormalities are observed after the bacteria colonize the soft tissues.

Our present research identified the same type of pathogenesis of vibriosis in Pacific oyster *C. gigas* larvae, as reported by Wang et al. 2021 [45]. Following the green fluorescence, we found that the GFP-tagged ME9 and NB10 strains were filtered through the velum and then entered into the stomach through the larval esophagus. Subsequently, by combining histopathological observations, it was clear that the pathogenic *Vibrio* proliferated rapidly in the stomach by 6 hpc and from that point onwards, it took 42 h to induced necrosis from larval digestive organs in the dorsal region to esophagus and velum. From then onwards, the destroyed larval tissues were expelled from the body. By 96 hpc, almost all soft tissues had been impaired with only few remnants left in the shell. These results are consistent with previous studies reporting that in the advanced stage of disease progression, bivalve larvae exhibited nearly complete necrosis of digestive organs [9,12,13,18,22,67].

The digestive system has been frequently reported as an infection pathway of vibriosis in bivalve larvae. Their attachment ability allows the *Vibrio* cells to grow along the digestive tract and destroy the digestive epithelium as a portal of entry [13,16,22,45]. Among the three types of pathogenesis of vibriosis reported by Elston and Leibovitz, two types of pathogenesis (type Ⅱ and Ⅲ) started from the digestive system [13]. Moreover, Sandlund et al. 2007 [16] observed *V. pectenicida* inside the digestive area and mucosal cells of great scallop *P. maximus* larvae through immunohistochemical staining. Dubert et al. 2016 [22] reported that GFP-tagged *Vibrio* strains were filtered by the larval velum and entered into the digestive system through the esophagus in Manila clam (*R. philippinarum*) larvae. Interestingly, larval velum, esophagus, stomach, and digestive glands were suggested to be the potential sites of the innate immunity system in bivalve larvae [35,68]. Many immune molecules, including C-type lectin, caspase-3, SOD, catalase, lysozyme, and LPS-binding protein, and bactericidal/permeability-increasing protein (LBP/BPI), were located in the digestive system of D-veliger bivalve larvae [35,68]. Therefore, a better understanding of the immune response against bacteria challenges, following the digestive system infection routes of vibriosis in bivalve larvae, appears to be particularly important.

In the present study, as compared with the unchallenged larvae, only the transcription levels of *IKK* and *NF-κB* that relate to signaling transduction were significantly up-regulated at 18 hpc. NF-κB is considered as a first responder and “rapid-acting” messenger to harmful cellular stimuli such as bacterial antigens [69]. The IKK/NF-κB signaling pathway was proven to induce the expression of immune-related genes in adult *C. gigas* [42,70,71] and proposed to function in *C. gigas* larval development [35]. As the infection was progressing, more immune-related genes were involved in the defense mechanism. The mRNA levels for *integrin β-1*, *peroxinectin,* and *HSP70*, were up-regulated at 30 hpc when the stomach and style sac had already been damaged. Cell adhesion is crucial in the cell-mediated immune response in invertebrates against foreign intruders [72]. Integrins, a family of cell adhesion molecules, play important roles in innate cellular immune responses, such as LPS binding activity, encapsulation, and phagocytosis in invertebrates [41,73]. In oyster *C. gigas*, integrin β-1, specially located in the velum of D-veliger and umbo larvae, was reported to mediate the phagocytosis towards *V. splendidus* with LPS binding activity [35]. Another essential cell adhesin molecule in invertebrates is peroxinectin, which functions in degranulation, encapsulation enhancement, opsonin, and peroxidase [72,74,75,76]. Isolated from crayfish *Pacifastacus leniusculus* and tiger shrimp *Penaeus monodon*, peroxinectin has been reported to have activities of cell adhesion and peroxidase in the presence of β-1,3-glucans and lipopolysaccharide (LPS) [38,77]. Moreover, the transcription of *peroxinectin* was up-regulated by *Vibrio alginolyticus* infection in white shrimp *Litopenaeus vannamei* [39] and by heat shock stress in Pacific oyster *C. gigas* adults [40]. Our study is the first to study peroxinectin in *C. gigas* larvae, and indicates that both integrin β-1 and peroxinectin may provide an adhesive/defensive activity in oyster larvae. Finally, HSP70s were frequently reported as typical stress-responsive proteins to provide cytoprotection to bivalve adults and larvae [25,35,37]. The results suggested that that this is also true for *C. gigas* larvae.

Bivalve larvae are more sensitive to pathogens since the immune system is not well developed in early stages of ontogenesis [24]. The immune system was suggested to be initiated during the trochophore stage of *C. gigas* larvae, while the early umbo larvae was the stage with a well-developed immune system [35]. In the presence of pathogens, the immune response occurred earlier as a defense against bacterial challenge and all the tested 12 immune genes were up-regulated significantly in D-veliger larvae of *C. gigas* after infection [35]. However, the high mortality observed in our study suggests that the immune system in *C. gigas* D-veliger larvae still did not provide efficient defensive activity after pathogenic *Vibrio* challenge. The delay in functioning of most of the tested immune genes may give a reasonable explanation as to why the pathogenic bacteria are able to impair tissues before the immune system response.

## 5. Conclusions

We firstly combined GFP-tagging, histological, and transcriptomic analyses to clarify in detail the pathogenesis of experimental vibriosis, and in reverse the mechanisms used by the host *C. gigas* larvae for disease resistance. The digestive system was verified as the infection pathway of *Vibrio* strains ME9-GFP and NB10-GFP in *C. gigas* larvae. In a hatchery, according to the infection pathway observed in our study, clinical signs such as abnormal swimming behavior could be regarded as symptoms synchronous with advanced vibriosis, since by the time the larvae show signs of infection, complete colonization has already occurred and curative measures will hardly be effective. Moreover, combining information on the transcription of immune genes with the infection process may be an essential tool to provide a better understanding of the response of immune genes to vibriosis. Most of the mRNA levels for essential immune genes in *C. gigas* larvae were not up-regulated until necrosis had already started in the digestive tissues. This suggests that the immunity in the early stages of *C. gigas* is not strong enough to prevent vibriosis, and future research may focus on strengthening of the gastrointestinal immune ability as a defense against vibriosis in bivalve larvae.

## Figures and Tables

**Figure 1 microorganisms-09-01523-f001:**
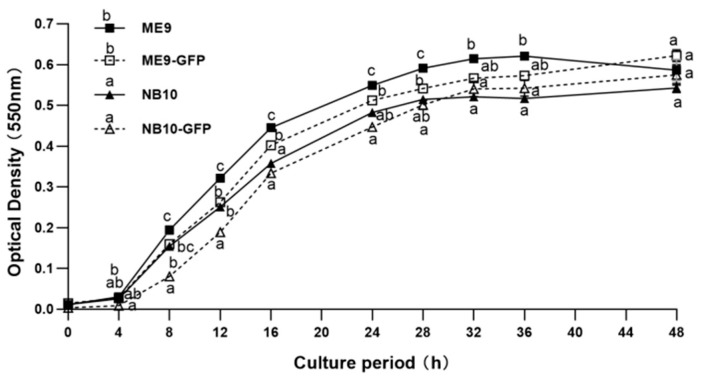
Growth of the parental *Vibrio* strains *V. splendidus* ME9, *V. anguillarum* NB10 and their respective GFP-tagged strains ME9-GFP and NB10-GFP, expressed as increase of optical density over a 48 h culture period. Data are expressed as mean ± S.E (*n* = 4). Values at each time point that are marked with a different letter are significantly different (*p* < 0.01, two-way repeated measures ANOVA).

**Figure 2 microorganisms-09-01523-f002:**
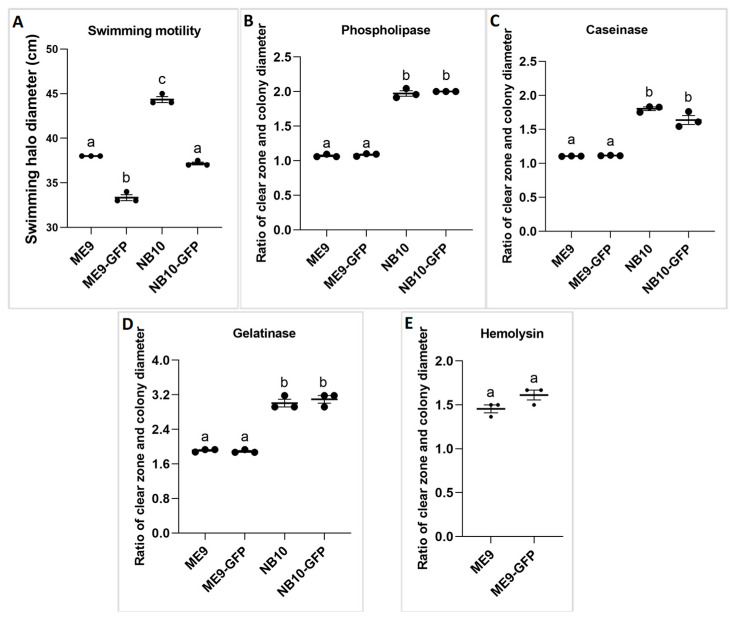
Activity of (**A**) swimming motility, (**B**) phospholipase, (**C**) caseinase, (**D**) gelatinase, (**E**) hemolysin of ME9, NB10, and their GFP-tagged strains. The swimming motility is expressed as the diameter of the motility zone. The activity of the other virulence factors is expressed as the ratio of the clear zone (mm) versus the colony diameter (mm). (**E**): Hemolytic activity was only observed in the strains ME9 and ME9-GFP while NB10 and NB10-GFP did not show this activity. Results are expressed as mean ± S.E (*n* = 3). Values that are marked with a different letter are significantly different (*p* < 0.001, one-way ANOVA).

**Figure 3 microorganisms-09-01523-f003:**
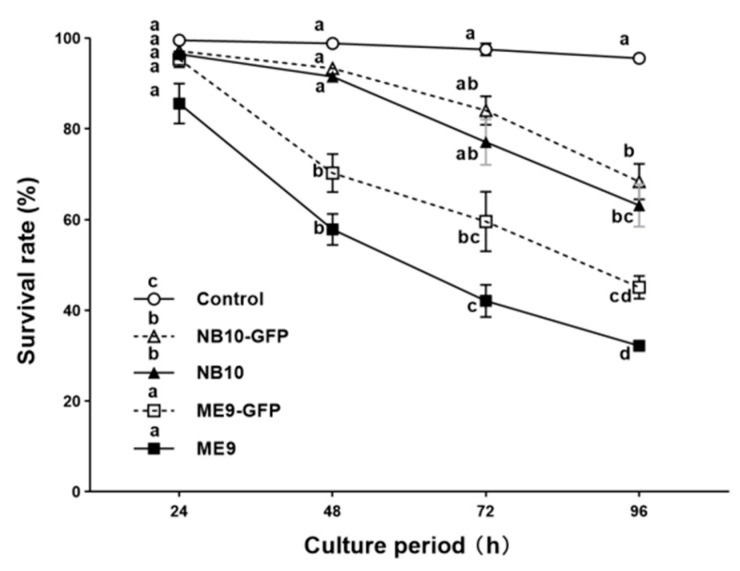
Experiment 1: Survival (%) of *C. gigas* larvae over 96 h exposure to *V. splendidus* ME9, *V. anguillarum* NB10 and their GFP-tagged strains ME9-GFP and NB10-GFP (10^6^ CFU mL^−1^). Data are expressed as mean ± S.E (*n* = 3). Values at each time point that are marked with a different letter are significantly different (*p* < 0.01, two-way repeated measures ANOVA).

**Figure 4 microorganisms-09-01523-f004:**
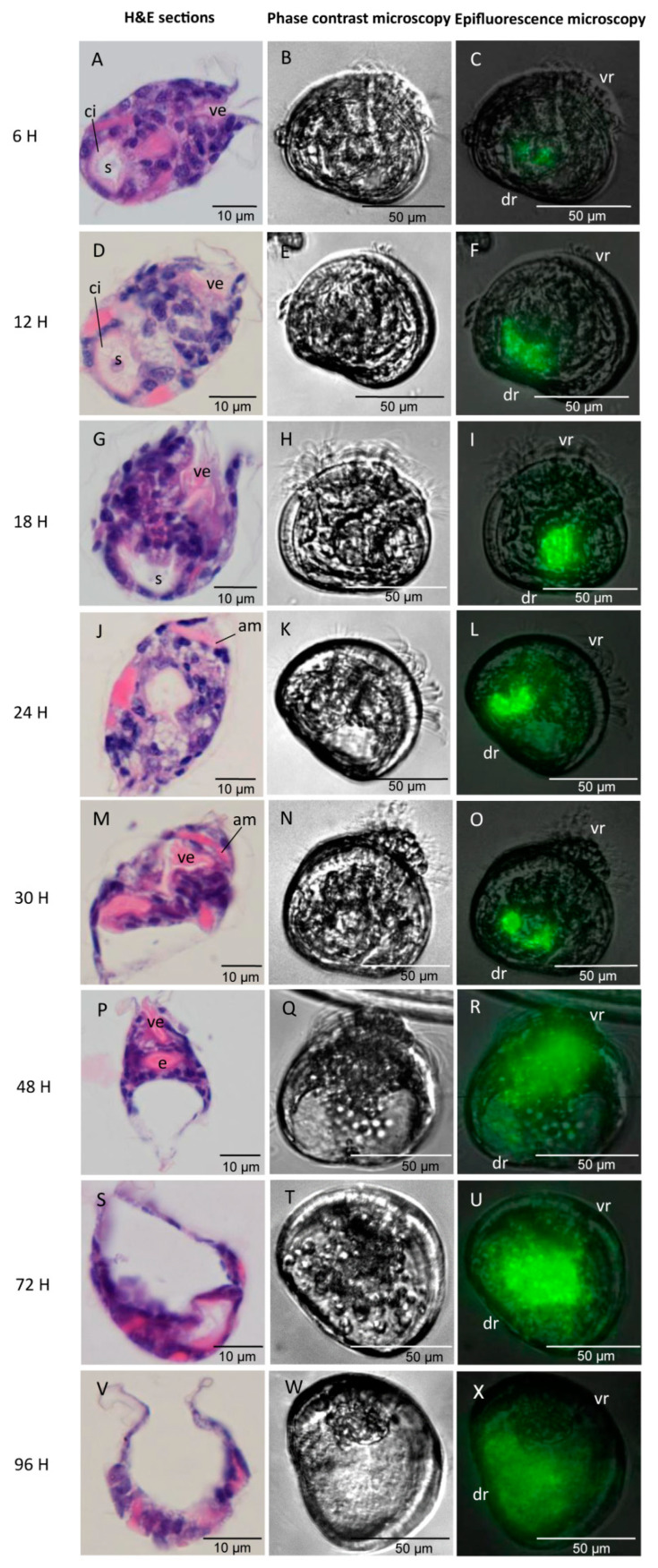
The invasion pathway of GFP-tagged *V. splendidus* ME9 and the synchronous histological damages in *C. gigas* larvae (**A**–**X**). Phase Ⅰ or incubation phase (6–24 hpc); Phase Ⅱ or rapid diffusion phase (30–48 hpc); Phase Ⅲ or acute mortality phase (72–96 hpc). The localization of ME9-GFP in larvae was determined by epifluorescence microscopy and the internal destruction following the infection process was confirmed by synchronous histological observation of H&E sections. The same larva was observed under phase contrast microscope and epifluorescence microscope. Abbreviations: am: anterior adductor muscle; ci: cilia; dr: dorsal region; e: esophagus; s: stomach; ve: velum with associated cilia; vr: ventral region.

**Figure 5 microorganisms-09-01523-f005:**
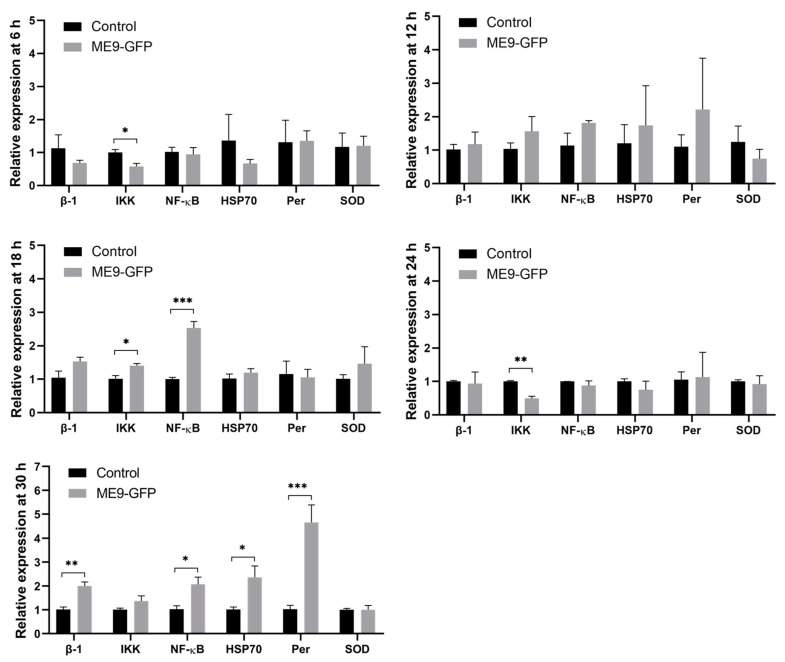
Temporal mRNA expression patterns of immune-related genes in *C. gigas* larvae challenged with *V. splendidus* ME9-GFP at 6, 12, 18, 24, and 30 h post challenge. Data are shown as mean ± S.E (*n* = 3). Asterisks represent significant difference between the control and the treatment: * *p* < 0.05, ** *p* < 0.01, *** *p* < 0.001, Student’s *t*-test.

**Table 1 microorganisms-09-01523-t001:** Primers used for reverse transcriptase qPCR of *C. gigas* larvae immune-related genes.

Genes ^a^	Gene Description ^b^	Primer Sequences (5′-3′)	Reference
*RS18*	Ribosomal protein S18	F: GCCATCAAGGGTATCGGTAGAC R: CTGCCTGTTAAGGAACCAGTCAG	[53]
*RL7*	Ribosomal protein L7	F: TCCCAAGCCAAGGAAGGTTATGC R: CAAAGCGTCCAAGGTGTTTCTCAA	
*SOD*	The antioxidant enzymes	F: TGAAGGCCGTCTGTGTATTG R: TCCATGCTGTCCTGGTGTTA	AJ496219 ^c^
*HSP70*	Acute phase proteins	F: CCAGTTGAGGATACTCTTGAGTGC R: ATGTCGATAACGGTCCCTTTCT	[35]
*IKK*	Immunity signaling pathways	F: TCTCACACCCACACACCTATGC R: AGTAGTTTTCCACCAGGGGATAAG	
*Rel*/*NF-**κ**B*		F: GAAGGCAAAGGGAGGTGATGAG R: GGTGTGCGGAAGACAATGGC	
*Integrin β-1*	Cell adhesion molecule	F: TCATCTGTGGAGGTCTGAGTCG R: TGTACATGCAGGGGCTTTTGTC	
*Peroxinectin*		F: GCCAAACCTCGCCTACCTTC R: GTGGAGTTGACGCGTGACATA	[40]

^a^: Gene symbols according to NCBI’s Gene Database. ^b^: According to UniProt Database. ^c^: Locus tag in the Crassostrea gigas genome sequence (GenBank).

## Data Availability

The authors declare that the data supporting the findings of this study are available within the article.

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
