# Peer review of "Dynamic Immune Response to Vibriosis in Pacific Oyster Crassostrea gigas Larvae during the Infection Process as Supported by Accurate Positioning of GFP-Tagged Vibrio Strains"

_microorganisms, 2021, doi:10.3390/microorganisms9071523_

Round 1
Reviewer 1 Report
The manuscript 1285983 is well written and its main focus well defines. The data obtained are valuable to define the dynamic of infection of C. gigas larvae by bacteria and starting to identify and characterized the immune response associated.
However, some weak in the description and in the statistical analysis of the results had been observed. This review suggested a improvement of the manuscript through a more objective description of the results and their subsequent statistical analysis.
2.4.2. The authors stay “The survival of larvae in each experiment was measured, but only the data from experiment 1 are shown here. The survival in other experiments was used to check if the toxicity of the tested strains remains stable on oyster larvae”.
In order to assess the stable dynamic of the toxicity on oyster larvae through the different experiment, is needed to show that all the experiments had a similar mortality rate so you can compared between them in terms of different obtaining results. The authors should show all the mortality rates of all the experiments at least in a table or so.
2.4.3. In experiment 2 descriptions. Were these larvae collected for sampling still alive? How do the authors to monitories and defer between death or alive larvae at each sampling point? Please explain.
2.4.4. Experiment 4. Did the authors sample only alive larvae, I guest? Did they? How they differ between alive or death larvae?
2.5.3.2. Did the authors design all the primers from the sequence os all the genes? or they used previously published primers for most of the genes and design only those which were not previously used in any publication?
How many primers did the authors design?
Second sentence of the paragraph: This sentence need revision, the RS18 and RL7 are housekeeping genes of larvae but not for bacterial strains. Did not they?
2.5.3.3. Here authors described a semiquantitative methodology. They did not perform a reverse transcriptase real-time PCR as they did not start from RNA. They performed a real-time PCR from cDNA.
Results, line 316. The authors did not explain the data regarding the swimming halo of the different bacteria in the text (Fig. 2A).
The GFP-bacteria showed differences in mobility versus their parental strains.
3.4. What about the immune response against the NB10-GFP bacteria. Both bacteria had the same colonization pattern, but that do not means that their ability of triggered the immune response will be the same. In fact, they are two different species of bacteria with quite different pathogenic activities and probably the immune response against them will be different.
Moreover, they have different pathogenic capabilities as described in the results at point 3.1.
Figure 5. Authors should name each graph of the same figure with letters n order to describe both in a concise manner in the text.
- Relative expression levels versus what?
Are the authors showing their data in fold change of the control levels?
Or are they showing the expression levels only in the infected specimens through infection time?
Why did they not perform any statistical analysis to assume differences between in the levels of expression between different time points?
- Are authors sure that there are statistically significant differences in the expression levels at the different time points. It seems not to be. The biological variation is too high in certain point. So probably the levels of SOD kept steady during the infection period.
Discussion
Line 471: This change (Figure 5) could be due to normal variation of immune homeostasis. If there are no differences in expression levels between infected and control larvae, authors cannot assume that the differences observed are due to infection. I do not understand the expression levels dynamics of infected larvae without comparing them with the control ones.
Sentence inline 477: Probably, but authors show a inhibition of IKK short after infection, and then a stimulation of IKK together with the up-regulation of NF-KB expression. Subsequently IKK is again down-regulated in infected larvae compared to control ant 24 hpi. Could the authors explain this and their relation with a impair immune response?
Regarding to the levels of immune genes discussion:
Most of the immune effectors genes analyzed are up-regulated too late. Probably this could explain why the bacteria are able to impair the tissues before the immune system response.
How the authors explain why NF-Kb is inducing at 18 hpi and 30 hpi, but not at 24 hpi?
Reviewer 2 Report
Review of the manuscript: ‘Dynamic Immune Response to Vibriosis in Pacific Oyster Crassostrea gigas Larvae during the Infection Process as Supported by Accurate Positioning of GFP-tagged Vibrio Strains’.
There are aspects of the technical presentation that require attention:
Make sure that all works cited in the text are in the reference list, that the presentation is consistent and that correct information is given.
Define and explain all acronyms and abbreviations on first mention in the text.
On first mention of a species in the text give both the common (trivial) and formal name, and make sure that the presentation is correct and consistent. For example, line 158.
Make sure that symbols, sub- and super-scripts, upper- and lower-case are presented correctly, and that there is correct and consistent use of italics, brackets and punctuation etc.
There are mistakes in the reference list, including incorrect reporting, inconsistent presentation, spelling mistakes and problems with use of punctuation etc.
